# Epidemiology of Status Epilepticus in Kazakhstan: A 10-Year Population-Based Study

**DOI:** 10.3390/jcm14248911

**Published:** 2025-12-17

**Authors:** Ruslan Akhmedullin, Temirgali Aimyshev, Zhasulan Utebekov, Gaziz Kyrgyzbay, Darkhan Kimadiev, Abduzhappar Gaipov

**Affiliations:** 1Department of Medicine, Nazarbayev University School of Medicine, Astana 010000, Kazakhstan; 2Epileptology Centre, Medical Center Hospital of the President’s Affairs Administration of the Republic of Kazakhstan, Astana 010000, Kazakhstan

**Keywords:** status epilepticus (SE), epidemiology, mortality, epilepsy, rescue medicines

## Abstract

**Background/Objectives:** This study explored the epidemiology of Status Epilepticus (SE) in Kazakhstan. **Methods:** Utilizing data from the National Health System from 2014 to 2023, we investigated the age-standardized incidence rate (ASIR) of SE. The authors employed restricted mean survival time (RMST) models to evaluate how sex, older age, epilepsy, history of cerebrovascular diseases (CVD), central nervous system (CNS) infections, brain tumors, and cancer affected survival during 30 days through the fifth year following hospital admission for SE. **Results:** This study included 14,010 patients. The ASIR per 100,000 increased threefold, from 4.15 (95% CI: 3.85; 4.46) in 2014 to 12.12 (95% CI: 11.64; 17.59) in 2023, with a sharp increase during the COVID-19 pandemic. The 30-day and 5-year mortality were 2.10% and 8.85%, respectively. The RMST identified that all-cause mortality was driven by elderly age, brain tumors, and cancer, where the difference in survival increased from one day at baseline to over a year by the fifth year. The effects of CVD, CNS infections, and sex on survival were substantially lower. However, epilepsy was associated with a better prognosis. **Conclusions:** We observed an incremental increase in the SE incidence over a decade. Our findings warrant actions to resolve issues related to rescue medicines to improve SE outcomes in both country and region. It may be a priority for elderly patients and those with systemic tumors. Further research is needed to understand the role of epilepsy in SE epidemiology, with emphasis on design-related biases.

## 1. Introduction

Status epilepticus (SE) represents a life-threatening neurological condition, requiring substantial medical resources [1]. Existing evidence suggests that the annual incidence of SE worldwide is up to 41 per 100,000 [2,3]. The burden of SE generally follows a bimodal distribution, with the highest estimates observed in children and the elderly and is associated with significant mortality [4].

SE is a clinical sign that reflects a wide range of etiologies [1]. A previous review [5] revealed notable differences in case fatality despite the similarities in the etiology of SE between developed and developing countries. However, a recently published study suggested that the etiology varies geographically and is often apparent less frequently, with notable challenges in identifying it [1]. Nevertheless, the existing literature indicates that developing regions face notable challenges in medical infrastructure [6,7], and such a discrepancy in mortality may be driven by poor healthcare resources [5].

Epidemiological studies that provide estimates of SE occurrence and burden are essential for healthcare planning. Currently, data on developing locations are limited, with major regions still being undiscovered [8,9]. Kazakhstan is one of five countries in Central Asia, where data on the burden of SE remain scarce. The latest Global Burden of Disease study (GBD) [10] reported that the country has the highest incidence rate in Central Asia. However, the study’s estimates combined data on both epilepsy and SE. Furthermore, the limited data sources used in GBD have traditionally been one of the major constraints [11].

To this end, we carried out this study to contribute to the data from low-resource locations by exploring health data from Kazakhstan, the largest database in the region. We start with exploratory statistics, then present model estimates for survival outcomes, and complete by discussing our findings and current SE-related challenges in a country.

## 2. Materials and Methods

### 2.1. Data Source and Study Design

This study used data from the Electronic Registry of Inpatients (ERI), which is part of the Unified National Electronic Health System (UNEHS) between 2014 and 2023, inclusively [12]. The UNEHS was launched in 2003 and established in 2014. The public healthcare sector, run by the Ministry of Health, covers approximately 70% of all healthcare facilities in Kazakhstan and provides medical services for the entire population. There is an extensive hospital network in the UNEHS database that increases extent over time. Although the UNEHS covers the majority of public hospitals, admissions registered within penitentiary and military systems were not available for analysis. However, a recent study reported mortality estimates of less than 8% when contrasting the estimates from the UNEHS with those reported by the local Bureau of National Statistics [13]. Hence, the ERI made medical claims available throughout the country. The authors conducted a retrospective longitudinal study to provide an age-standardized incidence rate (ASIR) and model survival in a cohort of patients hospitalized due to SE over a 10-year period.

### 2.2. Study Population

We included patients of all ages with a primary diagnosis of SE, specified by the International Statistical Classification of Diseases, Tenth Revision (ICD-10), G41. The local guidelines for the diagnosis and treatment of SE use the latest ILAE definitions [1].

### 2.3. Variables

Individual patient data encompassed a unique population registry number (RPN ID) for a patient, with additional information such as sex, age, date of birth, date of admission(s), coexisting epilepsy, history of cerebrovascular diseases (CVD), central nervous system (CNS) infections, brain tumor, cancer, and date of death with a corresponding cause (ICD-10) linked to the death certificate, if applicable. All data were complete, with no missing values observed. All corresponding ICD—10 codes are available in the Appendix A.

### 2.4. Measures

We provided incidence estimates for the first episode of SE and overall hospital admissions for SE. Incident SE was defined as the first SE hospitalization during the study period, and the overall incidence represented acknowledging the yearly admission counts. The authors also incorporated data on (all-cause) mortality occurring within 30 days and deaths recorded during the following years (i.e., 1, 2, 3, 4, and 5).

### 2.5. Follow-Up and Outcome Assessment

The entry date was the date of first admission, and the endpoint of the follow-up period was as 31 December 2023. The censoring date was defined as the date of death for those who died or 31 December 2023, otherwise. Survival time was calculated as the difference (in days) between the entry and censoring dates of the patients.

### 2.6. Statistical Analysis

Data cleaning and analyses were performed using the R (version 4.5.1) programming environment. Data were summarized as frequency and percentage. *p* values were two-sided at the conventional (5%) significance level.

#### 2.6.1. Incidence of SE

The authors used the total population size of Kazakhstan from 2014 to 2023, which was obtained from the local Bureau of National Statistics, and the incidence rates were standardized using the World Health Organization’s standard population distribution [14]. We then used the “surveil” package to calculate the ASIR and annual percent change (APC) for both incidence estimates. The rates were expressed per 100,000 population.

#### 2.6.2. Survival Analysis

Cox regression models have been the primary choice for “time-to-event” data. This requires the hazard ratio (HR) to be proportional over time. However, it is rarely held in practice, especially when the effect (naturally) changes over time [15], which was evident in our data. For instance, the HR fluctuated over time for epilepsy and a history of CVD during the study period (Appendix A). To address this issue and propose easily interpretable estimates, we employed a restricted mean survival time (RMST) analysis [16,17]. Briefly, the RMST relaxes statistical assumptions, remains less sensitive to the frequency of events, and provides a robust measure of the difference in survival. It calculates the area under the survival curve for a specific time point and can be interpreted as the average survival time at that point. As RMST requires specifications of truncation time points, we report both the difference in the average survival time and its ratio for 30 days, as well as for the following years from 1 to 5. This study was performed in accordance with the “STROBE” guidelines [18].

## 3. Results

Our study identified 14,010 patients with at least one SE-related hospital admission from 2014 to 2023 (Table 1). The cohort mostly consisted of males (54.16%) and those aged 20–64 (57.69%). Although the number of hospitalizations for SE per patient ranged from 1 to 25, the majority of patients were admitted once (79.54%) (Appendix A). As for comorbidities, patients mostly had a diagnosis of epilepsy (34.78%), a history of CVD (7.36%), cancer (2.46%), CNS infections (2.02%), and brain tumors (1.82%) (Table 1).

The ASIR for the first episode increased from 3.90 (95% CI: 3.61; 4.20) in 2014 to 7.20 (95% CI: 6.83; 7.57) in 2023 (Appendix A). Similarly, the overall ASIR (total admission counts) increased from 4.15 (95% CI: 3.85; 4.46) in 2014 to 12.12 (95% CI: 11.64; 12.62) in 2023. Both indices saw a sharp increase in 2021, which decreased in the following years. The corresponding APC in 2021 was 59.78 (95% CI: 49.82; 70.43) and 55.54 (95% CI: 47.42; 64.08), respectively. Despite the observed fluctuations, on average, ASIR (first episode) over the decade increased by 11.10% (95% CI: −10.53%; 32.73%), while ASIR (total admissions) rose by 16.96% (95% CI: −6.33%; 40.26%).

During the study period, 1376 (9.82%) deaths were observed. Among them, the leading assigned causes were “G93” (245, 17.81%), followed by “G41” (149, 10.83%), “G40” (136, 9.88%), “C71” (46, 3.34%) and “I63” (42, 3.05%), cumulatively covering 45% of all death causes. Mortality increased from 2.07% at 30-day to 8.85% in the fifth year. When modeling data using multivariable RMST across time points, the strongest predictors of all-cause mortality were elderly age, brain tumors, and cancer. At 30 days, the survival difference between the groups was modest, approximately one day. However, this gap increased progressively each year, reaching its peak at five years of follow-up, with the largest reduction in survival being observed among individuals with cancer (441.7 days), followed by the elderly age (408.7 days) and those with a history of brain tumors (328.2 days). Male sex, a history of CVD, and CNS infections were also associated with reduced survival, although to a less pronounced (Figure 1, Appendix A). In contrast, the presence of comorbid epilepsy was associated with improved survival over time compared with that in individuals without epilepsy (Figure 1, Appendix A).

## 4. Discussion

Our study found a nearly three-fold increase in hospitalizations for SE over the last decade. There was a marked surge in 2021, which was followed by a decrease thereafter. The most important covariates for both short- and long-term survival in the SE cohort were comorbid cancer, brain tumors, and elderly age. In contrast, coexisting epilepsy was associated with a better prognosis.

Seizure-related emergencies are among the most common reasons for hospital admission [19]. The incidence estimates in our study showed a steady increase. Although the ASIR of 12.12 cases per 100,000 population in 2023 overlaps with existing epidemiological findings [2,5], the published results remain discordant. The ILAE suggested definitions and differences in methods of case ascertainment are an explanation [20]. For instance, although local guidelines have been developed respecting the latest ILAE operational definitions [1], diagnostic ascertainment remains a concern due to prolonged pre-hospital delays. In a recent study [21], when adjusting for the duration of SE, the incidence estimates halved. This may also be true for our study, which included only patients who experienced prolonged seizures or episodes that persisted until emergency services arrived or until hospital admission. Hence, our findings may underestimate the incidence of SE and should be interpreted cautiously. Improving coding practices and case definitions in future studies can affect epidemiological parameters. In addition, estimates of incident SE largely depend on the distribution of etiological factors [2,5]. Kazakhstan is among the countries with the highest burden of stroke worldwide [11]. Hence, acknowledging the high mortality rate of stroke in the country, patients with stroke-related SE might die before reaching a hospital, thereby lowering the incidence in our study. This pattern may also contribute to the underrepresentation of elderly (≥60) in our cohort, with their proportion being lower than previously reported [4]. Finally, previous studies have suggested that Kazakhstan is among the countries with the highest figures for idiopathic generalized epilepsy in the region [22,23], which is often associated with nonconvulsive SE [2]. Given the limited and disproportionate access to medical technologies in the country [24], timely and accurate SE diagnosis remains challenging, potentially affecting our findings.

There were sharp increases observed in the ASIR values in 2015 and 2021. The increase in 2015 is likely associated with the widespread implementation of the UNEHS across the country, which led to more medical claims being available in the database. After 2015, the ASIR estimates remained somewhat stable until 2021. However, the spike observed in 2021 may reflect the impact of the COVID-19 pandemic. A recent study found the highest mortality indices in 2021 [13], with ischemic heart disease, CVD, COPD, and brain and neuropsychiatric disorders emerging as the leading causes of death in Kazakhstan. We have no supportive data for this, but we speculate that the observed increase in SE admissions during the pandemic years might reflect the rise in the incidence of potential causes of SE [2] and the COVID-19 infection, itself [25].

The mortality rate following SE episodes varies widely in the literature, ranging from 0% to 22% in children and from 0% to 57% in adults [4,5,26]. Our mortality estimates and distribution of comorbidities were comparable to those reported previously [4,5,26,27]. In the survival analysis, elderly age, brain tumors, and cancer had a pronounced impact on mortality among patients with SE. Although the impact of the former is consistently reported in the literature [2], claims on the latter remain limited and even contradictory. Earlier work exploring a nationwide database identified cancer as having a detrimental effect [27], while the following study did not find statistical evidence [28]. Nevertheless, patients with cancer may have metastatic involvement of the nervous system or cerebral metastasis [29], leading to neurological complications, including SE. Tumor-related SE is associated with increased mortality compared to SE caused by other factors [30]. Hence, this may have clinical implications, suggesting that individuals with tumors may benefit from SE preventive strategies.

Existing evidence suggests that most SE episodes develop without pre-existing epilepsy [2,31]. Similarly, 34.78% of our cohort had an epilepsy diagnosis, which was associated with a slightly improved survival. Previous studies have also observed inverse patterns in the SE cohort [28,32]. It has been suggested that individuals with epilepsy may have regular medical monitoring to improve the management of the etiological factors of SE and awareness of its external triggers [32]. Additionally, patients without epilepsy tend to be younger and may debut with refractory SE, further deteriorating their prognosis [28]. It is also likely that collider (design) bias contributed to the counterintuitive estimates. It might be unintentionally introduced when controlling for a variable (e.g., hospital admission), further revealing erroneous findings. By analyzing only those who survived until hospital admission, we conditioned on a variable that could be influenced by both the severity SE and pre-hospital mortality. While this also involves a form of survivorship (selection) bias, the distortion in the observed associations might arise from the collider structure of the data, potentially contributing to counterintuitive associations. This highlights the importance of considering design-related biases when interpreting findings because of the restrictions imposed at the design, data collection, or analysis stage. Researchers should discuss all possible scenarios in which such an artifact could have occurred to lessen its impact when drawing inferences. Hence, the protective effect of epilepsy remains multifactorial and requires further investigation.

Seizure duration remains the only modifiable factor affecting SE outcomes [9]. Early diagnosis and timely initiation of effective treatment are essential [6,8]. The increased duration may result from the absence of treatment before hospitalization, inadequate treatment at admission, and/or decreased sensitivity to benzodiazepines (BZDs) [6,7]. The unidentified SE remains common during prehospital care and is associated with pronounced delays in first-line therapy [33,34]. Evidence suggests that the treatment of SE in resource-poor countries is notably complicated by the lack of rescue medicines [7,35]. Although various BZDs are recommended worldwide for different routes of administration [36], in Kazakhstan (like in either Central Asian country), according to the living guidelines on SE, the only available BZD is intravenous diazepam. At present, the use of BZD is under strict government control and is carried out only in healthcare organizations that have a license for trafficking psychotropic substances and precursors. However, most seizures occur outside hospitals, which precludes the timely administration of diazepam unless injected by emergency medicine professionals. Such a restriction might contribute to the increasing rates of hospital admissions for SE and affect both short-and long-term outcomes. Hence, expanding the list of rescue medicines and non-intravenous administration routes is necessary. Addressing these issues is crucial, highlighting the need for action from international foundations and local regulatory bodies.

## 5. Limitations

There are several limitations that should be considered when interpreting the findings of this study. First, we included only patients hospitalized for SE. This restriction may have introduced selection bias. Conditioning on hospitalization might lead to underrepresentation of the study population, since hospital admissions were only possible for those who survived an acute cause preceding hospital admission or those whose emergencies were unresolved. This could affect both incidence and mortality estimates, since incident cases were the denominator for mortality. By including only patients admitted alive or with unresolved episodes, the denominator for mortality and the numerator for incidence are both artificially reduced. It would be interesting to assess whether additional data on those who died prior to hospitalization or were discharged without hospitalization could affect the findings. Second, we focused on primary SE, setting aside those with secondary SE. Given that SE is a symptom rather than a disease, future studies should include patients with comorbid SE. Third, case definition and coding practices might matter, implying that some patients in our cohort might have had functional seizures and were wrongly classified as SE. Finally, in the survival analysis, we did not have data on other clinically important variables (e.g., duration of SE, refractoriness, and etiology), which could confound the observed associations. Future studies should expand the list of covariates that would improve the quality of inferences.

## 6. Conclusions

This was a pioneering study on the burden of SE in Central Asia. We observed age standardized incidence hospitalization rates increased threefold over a decade, with sharp increase in the COVID-19 pandemic. Older age and tumors returned the most essential factors of mortality in SE cohort, whereas coexisting epilepsy was associated with better prognosis. Expanding the list of available rescue medicines remains a priority for improving SE outcomes in both country and region.

## Figures and Tables

**Figure 1 jcm-14-08911-f001:**
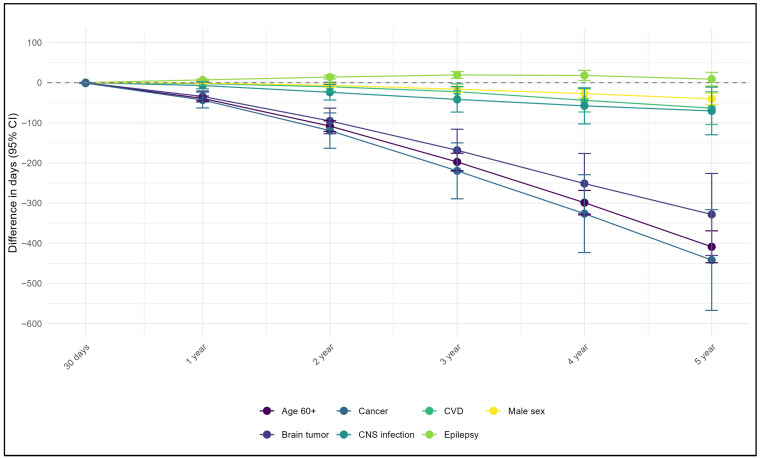
Difference in survival time from multivariable analyses using restricted mean survival time method.

**Table 1 jcm-14-08911-t001:** Baseline characteristics of the study cohort.

Covariate	Total, *n =* 14,010	30-Day Mortality, *n =* 290	1-Year Mortality, *n =* 686	2-Year Mortality, *n =* 911	3-Year Mortality, *n =* 1073	4-Year Mortality, *n =* 1160	5-Year Mortality, *n =* 1240	Overall Mortality, *n =* 1376
**Age group**								
0–4	2207 (15.75%)	18 (6.21%)	35 (5.10%)	39 (4.28%)	43 (4.01%)	43 (3.71%)	43 (3.47%)	43 (3.13%)
5–19	2778 (19.83%)	12 (4.14%)	24 (3.50%)	27 (2.96%)	30 (2.80%)	30 (2.59%)	30 (2.42%)	32 (2.33%)
20–64	8082 (57.69%)	178 (61.38%)	413 (60.20%)	552 (60.59%)	664 (61.88%)	730 (62.93%)	791 (63.79%)	898 (65.26%)
≥64	943 (6.73%)	82 (28.28%)	214 (31.20%)	293 (32.16%)	336 (31.31%)	357 (30.78%)	376 (30.32%)	403 (29.29%)
**Sex**								
Female	6422 (45.84%)	122 (42.07%)	275 (40.09%)	340 (37.32%)	395 (36.81%)	425 (36.64%)	449 (36.21%)	501 (36.41%)
Male	7588 (54.16%)	168 (57.93%)	411 (59.91%)	571 (62.68%)	678 (63.19%)	735 (63.36%)	791 (63.79%)	875 (63.59%)
**Epilepsy**								
No	9137 (65.22%)	228 (78.62%)	481 (70.12%)	610 (66.96%)	688 (64.12%)	736 (63.45%)	769 (62.02%)	829 (60.25%)
Yes	4873 (34.78%)	62 (21.38%)	205 (29.88%)	301 (33.04%)	385 (35.88%)	424 (36.55%)	471 (37.98%)	547 (39.75%)
**Cerebrovascular diseases**								
No	12,979 (92.64%)	254 (87.59%)	599 (87.32%)	784 (86.06%)	921 (85.83%)	992 (85.52%)	1065 (85.89%)	1184 (86.05%)
Yes	1031 (7.36%)	36 (12.41%)	87 (12.68%)	127 (13.94%)	152 (14.17%)	168 (14.48%)	175 (14.11%)	192 (13.95%)
**Central nervous system infection**								
No	13,727 (97.98%)	284 (97.93%)	666 (97.08%)	884 (97.04%)	1043 (97.20%)	1129 (97.33%)	1207 (97.34%)	1339 (97.31%)
Yes	283 (2.02%)	6 (2.07%)	20 (2.92%)	27 (2.96%)	30 (2.80%)	31 (2.67%)	33 (2.66%)	37 (2.69%)
**Brain tumor**								
No	13,755 (98.18%)	280 (96.55%)	635 (92.57%)	847 (92.97%)	1000 (93.20%)	1082 (96.94%)	1158 (93.39%)	1288 (93.60%)
Yes	255 (1.82%)	10 (3.45%)	51 (7.43%)	64 (7.03%)	73 (6.80%)	78 (3.06%)	82 (6.61%)	88 (6.40%)
**Cancer**								
No	13,832 (98.73%)	272 (93.79%)	636 (92.71%)	846 (92.86%)	998 (93.01%)	1079 (93.02%)	1155 (87.50%)	1277 (92.81%)
Yes	178 (1.27%)	18 (6.21%)	50 (7.29%)	65 (7.14%)	75 (6.99%)	81 (6.98%)	85 (12.50%)	99 (7.19%)
**Total hospital admission counts for SE**								
Once	11,143 (79.54%)	285 (98.28%)	626 (91.25%)	797 (87.49%)	916 (85.37%)	981 (84.57%)	1039 (83.79%)	1124 (81.69%)
More than once	2867 (20.46%)	5 (1.72%)	60 (8.75%)	114 (12.51%)	157 (14.63%)	179 (15.43%)	201 (16.21%)	252 (18.31%)

## Data Availability

The datasets used and/or analyzed during the current study are available from the corresponding author on reasonable request. The data are not publicly available due to restrictions from the Republican Center for Electronic Health of the Ministry of Health of the Republic of Kazakhstan, the Ministry of Health of the Republic of Kazakhstan.

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
