# Peer review of "Epidemiology of Status Epilepticus in Kazakhstan: A 10-Year Population-Based Study"

_jcm, 2025, doi:10.3390/jcm14248911_

Round 1
Reviewer 1 Report
Comments and Suggestions for Authors
Dear Authors,
The manuscript provides an interesting description of the issues related to the management of status epilepticus in a developing region.
Some clarifications could improve understanding of the data reported in the manuscript and in the supplementary materials.
Does the Electronic Registry of Inpatients include data from all hospitals in Kazakhstan or only some of them (included in 70% of all healthcare facilities)? If coverage is not complete, has it remained constant over the 10-year period studied?
For which population were the incidence data calculated? For the entire population of Kazakhstan, or for the catchment area of ​​the hospitals included in the database used?
A sharp increase is reported in the ASIR in 2021. However, an even greater percentage increase occurred in 2015 compared to 2014 (Table S3). A comment on this is appropriate.
A brief comparison between the incidence rates found in your study and those reported in the literature is appropriate; accompanied by some hypotheses on the origins of the discrepancy between the two data sets (and on the threefold increase in Kazakhstan in the decade studied).
The introduction notes that "the burden of SE generally follows a bimodal distribution, with the highest estimates observed in children and the elderly." The data presented in the manuscript contrast with this epidemiological curve, since the majority of cases (57.7%) occur in the 20-64 age group. This discrepancy also deserves comment.
Kind regards.
Author Response
Comment 1. Does the Electronic Registry of Inpatients include data from all hospitals in Kazakhstan or only some of them (included in 70% of all healthcare facilities)? If coverage is not complete, has it remained constant over the 10-year period studied?
Response 1: Thank you for bringing this to our attention. We appreciate your suggestion to strengthen the manuscript. To enhance the clarity, we have added a statement into “Data source and study design” section as follows: The public healthcare sector, run by the Ministry of Health, covers approximately 70% of all healthcare facilities in Kazakhstan and provides medical services for the entire population. There is an extensive hospital network in the UNEHS database that increases extent over time. Although the UNEHS covers the majority of public hospitals, admissions registered within penitentiary and military systems were not available for analysis. However, a recent study reported mortality estimates of less than 8% when contrasting the estimates from the UNEHS with those reported by the local Bureau of National Statistics. All the changes are highlighted in yellow. We thank you for your attention to this matter and hope that you appreciate our perspective on this matter.
Comment 2. For which population were the incidence data calculated? For the entire population of Kazakhstan, or for the catchment area of ​​the hospitals included in the database used?
Response 2: Apologies for confusion. To enhance the transparency, we have updated the corresponding statement as follows: The authors used the total population size of Kazakhstan from 2014 to 2023, which was obtained from the local Bureau of National Statistics, and the incidence rates were standardized using the World Health Organization’s standard population distribution. All the changes are highlighted in yellow for your convenience.
Comment 3. A sharp increase is reported in the ASIR in 2021. However, an even greater percentage increase occurred in 2015 compared to 2014 (Table S3). A comment on this is appropriate.
Response 3: We greatly appreciate your suggestion. To address this point, we have expanded the discussion. Please be informed that the following statement has been added: There were sharp increases observed in the ASIR values in 2015 and 2021. The increase in 2015 is likely associated with the widespread implementation of the UNEHS across the country, which led to more medical claims being available in the database. After 2015, the ASIR estimates remained somewhat stable until 2021. To make it easier for you to find the corrections, all the changes have been highlighted in yellow for your convenience.
Comment 4. A brief comparison between the incidence rates found in your study and those reported in the literature is appropriate; accompanied by some hypotheses on the origins of the discrepancy between the two data sets (and on the threefold increase in Kazakhstan in the decade studied).
Response 4: Thank you very much for this thoughtful comment. Please be informed that the following statement has been added to expand the discussion around our estimates: The incidence estimates in our study showed a steady increase. Although the ASIR of 12.12 cases per 100,000 population in 2023 overlaps with existing epidemiological findings [2,6,21], the published results remain discordant. The ILAE suggested definitions and differences in methods of case ascertainment are an explanation [22]… To make it easier for you to find the corrections, all the changes have been highlighted in yellow for your convenience.
Comment 5. The introduction notes that "the burden of SE generally follows a bimodal distribution, with the highest estimates observed in children and the elderly." The data presented in the manuscript contrast with this epidemiological curve, since the majority of cases (57.7%) occur in the 20-64 age group. This discrepancy also deserves comment.
Response 5: Thank you very much for pointing this out. Indeed, the proportion of elderly (aged 60 and more) was underrepresented in our study; however, the share of children was comparable to previous findings. This may be related to the high mortality rate of stroke in the country, which is one of the main causes of SE. Please be kindly informed that we have updated discussion as follows: Kazakhstan is among the countries with the highest burden of stroke worldwide [24]. Hence, acknowledging the high mortality rate of stroke in the country, patients with stroke-related SE might die before reaching a hospital, thereby lowering the incidence in our study. This pattern may also contribute to the underrepresentation of elderly (≥ 60) in our cohort, with their proportion being lower than previously reported [4,25]. All the changes have been highlighted in yellow for your convenience.
Reviewer 2 Report
Comments and Suggestions for Authors
The main question for the current manuscript is it significance for the general literature. It has undoubtedly a significant value for the authors own country, but its significance worldwide remains limited. Besides this, my comments are:
The title should include the type of study, country/region origin.
Abstract needs improvement. “largest country in Central Asia” is not scientific; add number of mortality rate, especially 30-day and 5-years, 30-day is worldwide acceptable.
Introduction. There is more recent data regarding SE and mortality, please provide evidence of the GBDs, or there are other sources that were published based in the CDC Wonder data. Add a paragraph about the country itself, what is unique about Kazakhstan regarding healthcare when compared globally and Asia.
Methods. Describe “How was missing data managed?” “How was ICD10 coding validated?” “How can be this data accessed, please add a link of the database”
Author Response
Comment 1. The title should include the type of study, country/region origin.
Response 1: Thank you very much for this suggestion. We fully agree that providing more information would be essential. Please be kindly informed that we have updated the title as follows: Epidemiology of Status Epilepticus in Kazakhstan: A 10-Year Population-Based Study. We hope you can appreciate our perspective on this matter.
Comment 2. Abstract needs improvement. “largest country in Central Asia” is not scientific; add number of mortality rate, especially 30-day and 5-years, 30-day is worldwide acceptable.
Response: Thank you bringing this to our attention. Please rest assured that the abstract has been updated by removing the mention of Central Asia and adding information on mortality. All the changes have been highlighted in yellow for your convenience.
Comment 3. Introduction. There is more recent data regarding SE and mortality, please provide evidence of the GBDs, or there are other sources that were published based in the CDC Wonder data. Add a paragraph about the country itself, what is unique about Kazakhstan regarding healthcare when compared globally and Asia.
Response 3: Thank you for your helpful suggestions. Please rest assured that we have updated the introduction as follows: Epidemiological studies that provide estimates of SE occurrence and burden are essential for healthcare planning. Currently, data on developing locations are limited, with major regions still being undiscovered [11,12]. Kazakhstan is one of five countries in Central Asia, where data on the burden of SE remain scarce. The latest Global Burden of Disease study (GBD) [13] reported that the country has the highest incidence rate in Central Asia. However, the study’s estimates combined data on both epilepsy and SE. Furthermore, the limited data sources used in GBD have traditionally been one of the major constraints [14]. All changes and updates are highlighted in yellow. We hope you can appreciate our perspective on this matter. Thank you once again for your valuable suggestion.
Comment 4. Methods. Describe “How was missing data managed?” “How was ICD10 coding validated?” “How can be this data accessed, please add a link of the database”.
Response 4: Thank you for highlighting this. For missing data, we mostly used administrative data, such as age, sex, ICD-10, and so forth. We were unable to obtain detailed clinical data. Therefore, our data were complete. Please rest assured that we have added the corresponding statement in the Methods section. For case identification, we used medical claims from the Electronic Registry of Inpatients, relying on the ICD codes. While these codes have not been validated, all procedures, including diagnosis confirmation, followed local guidelines, which require necessary medical tests for diagnosis verification. While some errors may exist, which were highlighted in the discussions and limitations, we still believe the data are valuable, especially considering the lack of data available for Central Asia. Regarding the access to the data, we expanded the Data Availability Statement as follows: The datasets used and/or analyzed during the current study are available from the corresponding author on reasonable request. The data are not publicly available due to restrictions from the Republican Center for Electronic Health of the Ministry of Health of the Republic of Kazakhstan, the Ministry of Health of the Republic of Kazakhstan.
Round 2
Reviewer 2 Report
Comments and Suggestions for Authors
Satisfactory